

# Pilates versus resistance training on trunk strength and balance adaptations in older women: a randomized controlled trial

María Carrasco-Poyatos[1], Domingo J. Ramos-Campo[2] and Jacobo A. Rubio-Arias[2]

[1] Department of Education, Health and Public Administration Research Center, Universidad de Almería, Almería, Spain
[2] Department of Physical Activity and Sport Sciences, UCAM Research Centre for High Performance Sport, Universidad Católica San Antonio, Murcia, Spain

Corresponding author
María Carrasco-Poyatos,
carrasco@ual.es

## ABSTRACT

**Background:** The neuromuscular decline impact in the functional independence of older women is determining the necessity of implementing new strategies focused on core strength training and postural stability maintenance to promote healthy aging.

**Objectives:** To define whether Pilates or resistance training is better at improving (a) core isometric and isokinetic muscular strength, and (b) static and dynamic balance, in older women.

**Methods:** This was a cluster randomized controlled trial. Physically independent older women (60–80 years) from day centers were randomly allocated to Pilates, Muscular and Control Groups (PG, MG and CG) using a block randomization method. Only the research staff performing the assessment and statistical analysis were blinded. Exercise groups trained twice a week (1 h per session) for 18 weeks in a moderate-to-vigorous intensity. Core strength (primary outcome): trunk and hip isometric and hip isokinetic muscular strength (Biodex System III Pro Isokinetic Dynamometer), alongside one leg static balance (portable force platform Kistler 9286AA) and dynamic balance (timed up and go (TUG)) were assessed.

**Results:** A total of 60 participants were randomized (PG, $n = 20$; MG, $n = 20$; CG, $n = 20$) and 49 completed the trial (PG, $n = 16$; MG, $n = 19$; CG, $n = 14$). Regarding hip isometric extension strength, PG was statistically better than CG ($P = 0.004$). There were no differences between groups regarding isokinetic strength or balance. Intra-group comparisons showed significant improvements ($P < 0.05$) in the dynamic balance and trunk and hip isometric extension strength for PG and MG, whereas every hip isokinetic measurement was improved in MG. Exercise programs did not produce any adverse event.

**Conclusions:** The Pilates training program was more effective for improving isometric hip and trunk extension strength, while the Muscular training program generated greater benefits on trunk and hip isokinetic strength. Moreover, both training programs showed moderate effects for the TUG.

**Clinical Trial Registration:** The trial was registered at ClinicalTrials.gov (identifier: NCT02506491).

## INTRODUCTION

The female gender is associated with lower odds of healthy aging with advancing age (*Rodriguez-Laso et al., 2018*). Due to their age-related hormone changes (i.e., menopause), women are more affected by this neuromuscular decline, which contributes to a worsening of functional independence and disability (*Newman et al., 2003*) and an increased risk of hospitalization and mortality (*Guadalupe-Grau et al., 2017*). Moreover, sarcopenia and muscle strength are negatively associated with balance and the risk and fear of falling in older women (*Gadelha et al., 2018*), thus falls and injuries are more frequent in women than in men (*Gioffrè-Florio et al., 2018*).

To reach the status of healthy aging, developing and maintaining functional ability that enables well-being is required. Thus, one of the primary objectives for functional maintenance in older women should be keeping postural stability (i.e., controlling the body's center of pressure) (*Horak, 2006*) and improving core strength, because research has shown a strong association between core strength and balance in the older generation (*Granacher et al., 2013*). In this way, the timed up and go (TUG) test is a quick way to determine the influential balance issues on elderlies' daily lives and for the prediction of future falls (*Sai et al., 2010*). In addition, low concentric muscle strength, assessed by isokinetic evaluation is the most accurate method to determine muscle activity (*Eyigor, Karapolat & Durmaz, 2007*) and low values of isometric strength have been associated with higher risk of falls (*Robinson et al., 2004*). Moreover, the decrease of the back muscle strength may lead to the quality of life decline and the falls increment in postmenopausal women with osteoporosis (*Miyakoshi et al., 2007*). Thus, the measurement of isokinetic and isometric hip and trunk strength can offer important information about physical factors related to healthy aging.

One of the most common types of exercise included in training for older people is multicomponent training as a combination of two or more of the following exercises: muscle resistance/strength, walking/endurance, balance and/or flexibility. Some systematic reviews and meta-analytical studies on this topic (*Marín-Cascales et al., 2018*) demonstrate a positive effect of strength training on cardiorespiratory fitness, body composition, metabolic outcomes, functional status, cognitive performance and quality of life in older people. Furthermore, a resistance training exercise program that focuses on the center of the body also results in positive effects on static (*Marques et al., 2011*) and dynamic balance (*Marques et al., 2011*; *Seo et al., 2012*) and improves the isokinetic strength of the knee (*Marques et al., 2011*).

Furthermore, during recent years a new type of training program called Pilates has been included as an effective method for improving physiological and psychological function. Some systematic reviews with meta-analysis showed strong evidence for Pilates training to improve static and dynamic balance (*De Souza et al., 2017*; *Moreno-Segura et al., 2018*) and lower limb strength, hip and lower back flexibility, and cardiovascular endurance (*De Souza et al., 2017*) in older adults. Moreover, studies involving older women indicate that Pilates-based exercise programs enhance isometric and isokinetic strength (*Irez et al., 2011*; *Bergamin et al., 2015*; *De Oliveira et al., 2017*; *Oliveira, Oliveira & De Pires-Oliveira, 2017*).

However, there is not enough evidence regarding the differences between two core exercise programs, such as resistance training or Pilates, on static or dynamic balance and core strength in this population to make the appropriate recommendations. Moreover, there is also a lack of information concerning core isometric or isokinetic muscular strength, as most studies have measured other corporal regions. For these reasons, the objectives of the present study were to determine what type of training creates greater adaptations in (a) core isometric and isokinetic muscular strength (primary outcomes), and (b) static and dynamic balance (secondary outcomes), in older women. Our hypothesis was that Pilates training would exacerbate increases in static and dynamic balance and isometric trunk and hip strength. We additionally hypothesized that resistance training would promote greater adaptations in isokinetic trunk and hip strength and dynamic balance.

## MATERIALS AND METHODS

### Design

This was a 18-week quasi-experimental randomized controlled trial in which independent older women were assigned to a Pilates Group (PG; $n = 20$), a Muscular Group (MG; $n = 20$) or a no-exercise Control Group (CG; $n = 20$). The trial was managed by the Faculty of Sport at San Antonio Catholic University (UCAM), Murcia, Spain, and was approved by the UCAM ethics committee. It was registered with ClinicalTrials.gov (NCT02506491, available from https://clinicaltrials.gov/show/NCT02506491), and the trial design followed Consort guidelines. Before starting the study and owing to an expert revision, original primary and secondary outcome measures were restructured in order to make the design more precise. This reorganization caused a delay in the beginning of the measurement date, starting on January and finishing on May (2016). Moreover, the final sample enrolled in the study was 60 instead of 80 women.

### Participants

A total of 80 older women (60–80 years) were invited to participate in the study. They were recruited from old people day centers from Murcia (Spain). These are centers were non institutionalized old people achieve activities such as painting, shewing or gardening. A general medical evaluation was accomplished to ensure they were physically and mentally able to participate in the exercise programs. It included the control of age, the level of education, toxic habits, medical treatment and/or diseases that can affect musculoskeletal or cardiovascular systems (self-report), mental illness –measured with the Mini-Mental state (*Folstein, Folstein & McHugh, 1975*), urinary incontinence, the presence of oedema and high blood pressure, and the independence to develop basic and instrumental activities of daily living, measured with Katz and Lawton and Brody scales (*Katz et al., 1963*; *Lawton & Brody, 1969*). Inclusion criteria were: women 60–80 years old who were physically able to develop the basic and instrumental activities of daily living and were without cognitive impairment or diseases that can affect musculoskeletal or cardiovascular systems. The exclusion criteria were: women who were currently participating or had previously participated in a structured Pilates or resistance training

exercise program in the past 3 months and those with a visual or auditory impairment not corrected with glasses or a hearing aid. Participants also had to maintain at least 80% (29 sessions) compliance with the exercise session. Sixteen women did not meet the inclusion criteria and four refused to participate. In total 60 women were actually enrolled in the study and randomly distributed into PG, MG and CG. All participants signed a consent form before the beginning of the study. Data were collected at the UCAM high-performance sport research center.

## Interventions

Participants allocated to PG or MG were required to train twice a week (1 h per session) for 18 weeks from January to May (2016). Women assigned to CG were encouraged to maintain their normal physical activity habits. The exercise programs were conducted by the same accredited exercise expert who was certified in personal training and Pilates.

The programs were divided into a 2-week familiarization period and four 4-week mesocycles that were designed to be progressively more challenging. An example of the training progression and the exercises implemented can be seen in Table 1. The sessions were given in three phases: (1) the warm-up, (2) the Pilates or resistance training exercise programs and (3) the cool-down. Intensity was controlled using the OMNI-Resistance Exercise Scale of perceived exertion (Robertson et al., 2003), beginning at a moderate intensity (six to seven points) and finishing at a moderate-to-vigorous intensity (eight to nine points).

The Pilates and resistance training exercise programs were focused on the spine, hip and girdle regions, stimulating the muscles in a dynamic and static way and exercising the arms and legs. Balance was an essential part of the standing exercises, and movements were always coordinated with breathing. In addition, the Pilates exercise program also incorporated the principles of Pilates, such as recruiting the body center's deep stabilizers to prepare movement, keeping the pelvis and the shoulder girdle in a neutral position to allow the extremities to disassociate from the trunk and being conscious of every aspect of all exercises to obtain correct and more valued movements. An example of Pilates exercises is presented in Table S1.

## Outcomes

The primary outcome measures were trunk and hip isometric and isokinetic strength. The secondary outcome was balance. The test was performed in all participants before and after the exercise intervention programs. The pre-tests were accomplished in January over a 1-week period.

### *Primary outcomes*

Core strength was determined by trunk and hip isometric (Tisom and Hisom) and hip isokinetic (Hisok) muscular flexion and extension strength. These parameters were assessed on a Biodex System III Pro Isokinetic Dynamometer (Biodex Medical System, Shirley, NY, USA). Before measurements were taken, participants were asked to warm up on a bicycle ergometer for 5 min using a self-chosen resistance of 40–60 rpm (20–30 watts), followed by 5 min of stretching exercises for the trunk and lower extremities

**Table 1  A total of 18 weeks training progression for Pilates and Muscular groups.**

| Mesocycle | Session example for Pilates group | Session example for Muscular group | Volume | Intensity | Density |
|---|---|---|---|---|---|
| Familiarization period (weeks 1–2) | General hip, spine and shoulders movilization recruiting body's center deep stabilizers | General hip, spine and shoulders movilization with transfer to the principal exercises | 4–6 repetitions/exercise | Breathing 1-1-1-1 (lower execution velocity)<br>No additional weight<br>OMNI-Res score of 4–6 points | Work/rest quotient of 1/4 |
| Mesocycle 1 (weeks 3–6) | Standing pelvic clock. standing spine twist. Standing hip extension. Hip abduction seated on a chair. Windmill arms seated on a chair. Standing floating arms. | Sitting and standing from a chair. Standing bent over row. Ankle flexion-extension grabbing the back of the chair. Arm side lateral. Standing push the partner for chest and biceps. Curl ups. | 6–8 repetitions/exercise | Breathing 1-1-1-1 (lower execution velocity)<br>No additional weight<br>OMNI-Res score of 6–7 points | Work/rest quotient of 1/2 |
| Mesocycle 2 (weeks 7–10) | Supine circle leg lifts. supine leg swing. Supine up shoulders with elastic band. Supine curl ups with chi ball. Side leg lifts. Standing shoulder circles with chi ball. Standing lateral flexion. | Squat grabbing the back of the chair. Standing bent over row. Ankle flexion-extension grabbing the back of the chair. Arm side lateral. Dumbbell press and biceps seated on a chair. Standing triceps. Curl ups. Elastic band for trunk and upper extremities exercises. | 8–10 repetitions/exercise | Breathing 1-1-1-1 (medium execution velocity)<br>Additional light-weight: elastic band<br>OMNI-Res score of 7–8 points | Work/rest quotient of 1/1.5 |
| Mesocycle 3 (weeks 11–14) | The bridge. Side leg lifts with chi ball. Prone hip extension. The cat with elastic band. Supine windmill arms with elastic band. Standing rolldowns. The hundred standing with elastic band. | Squat. Standing bent over row. Lunges. Arm side lateral. Dumbbell press and biceps seated on a chair. Standing triceps. Curl ups. Elastic band for trunk and upper extremities exercises. | 10–12 repetitions/exercise | Breathing 1-1 (higher execution velocity)<br>Additional moderate-weight: elastic band<br>OMNI-Res score of 8–9 points | Work/rest quotient of 1/1 |
| MESOCYCLE 4 (WEEKS 15-18) | Combining femur arcs and windmill arms. Pelvic curl with elastic band. Combining curl ups and shoulder abduction with elastic band. Side leg kicks. diamond press. Assisted roll up with elastic band. | Squat and front arms. standing bent over row. Lunges and up arms. Arm side lateral. Dumbbell press and biceps seated on a chair. Standing triceps. Curl ups. Elastic band for trunk and upper extremities exercises. | 12 repetitions/exercise<br>Combining upper and lower body exercises | Breathing 1-1 (higher execution velocity)<br>Additional moderate-weight: elastic band<br>OMNI-Res score of 9 points | Work/rest quotient of 1/0.5 |

Note:
OMIN-Res, OMNI-Resistance Exercise Scale of perceived exertion; Breathing 1-1-1-1, inhale to prepare the movement-exhale to go to the final position-inhale in the final position-exhale to go back to initial position. Breathing 1-1, inhale to prepare and go to the final position- exhale to go back to initial position.

(*Steinhilber et al., 2011*). Isokinetic testing was performed before isometric testing. For Hisok and Hisom assessments, participants lay supine on the dynamometer chair (*Meyer et al., 2013*). The chest, pelvis and non-tested thigh were fixed to the dynamometer chair using straps, therefore only the dominant side was assessed. The rotation axis was set at the level of the femoral joint (*Meyer et al., 2013*). For Hisok, the range of movement in the tested hip was adapted to the flexion capacity of each participant. For Hisom, the hip was fixed at 90° flexion. For Tisom assessment, participants were fixed in a standardized position (*Sekendiz et al., 2007*) with the trunk fixed at 90° flexion.

The rotation axis was set at the level of L5–S1 (*Ester et al., 2004*). For isokinetic testing, participants executed five concentric-concentric contractions at low (60°/s) and high (120°/s) velocity with 2 min of rest in-between. Prior to the test, a familiarization set of five submaximal repetitions was performed at each protocol speed. Following *Steinhilber et al. (2011)* and *Meyer et al. (2013)* for isometric testing, five sustained maximal voluntary isometric flexion and extension contractions of 5 s were executed with a 5-s rest period in-between. The parameters evaluated included peak trunk and hip isometric flexion and extension relative to weight (Tisom_Flw, Tisom_Exw, Hisom_Flw and Hisom_Exw), and also peak hip isokinetic flexion and extension at 60°/s and 120°/s relative to weight (Hisok_Fl60w, Hisok_Fl120w, Hisok_Ex60w and Hisok_Ex120w).

### Secondary outcomes

Static balance (SB) was implemented by one leg test under single-task conditions and was assessed using a portable force platform (Kistler 9286AA; Kistler instrumente AG, Winterthur, Switzerland). The signal was transmitted to a computer at a sampling rate of 100 Hz. The data were exported and processed in Excel (Microsoft Excel 2018 for Windows). Since there is no gold standard measure of balance (*Heyward, 2002*), the most common single leg SB protocol was implemented. Participants were barefoot and maintained an upright position with their hands hanging loosely down and their eyes open. Their gaze was fixed on a mark at eye level. Right and left single support was performed. The time (seconds) that they maintained the static position was measured. The displacement velocity of the center of pressure in the medio-lateral and antero-posterior planes, as well as the velocity moment, were calculated using the formula described elsewhere (*Ishizaki et al., 2002*). The mean of the right and left support was calculated for the data analysis. Variables were: SB_Time (s), SB_Vml (mm/s), SB_Vap (mm/s), SB_Varea ($mm/s^2$). Measurements were conducted in three 30-s trials with 1 min of rest in-between. Dynamic balance was assessed using the 3-m walk TUG test (*Rikli & Jones, 2001*). Participants were given one TUG familiarization trial followed by two maximal trials in a fast velocity. The best time was used in the analyzes.

## Sample size and power

Calculations to establish sample size were performed using Rstudio 3.15.0 software. The significance level was set at $\alpha = 0.05$. According to the standard deviation (SD) established for isometric trunk extension in a previous study (*Markovic et al., 2015*) and an estimated error (*d*) of 23 N/m, a valid sample size for a confidence interval (CI) of 95% was 46 ($n = CI^2 \times d^2/SD^2$). A total of 49 women completed the trial. The final sample size for each group obtained in our study (PG = 16, MG = 19, CG = 14) will provide powers of 78%, 85% and 69% respectively if between and within a variance of 1.

## Randomisation and blinding

A block randomization method was used to allocate participants to the groups with equal sample sizes (PG, MG and CG, $n = 20$). This randomization method was chosen according to allocation of the specialized senior centers. Block size was determined by the research staff according to the statistical power provided. Blocks were chosen randomly to

determine the participants' assignment into the groups. Following *Kim & Shin (2014)*, a randomization sequence was created using Excel 2016 (Microsoft, Redmond, WA, USA) with a 1:1 allocation using a random number table by one of the research staff member specialist in statistical analysis. Owing to the difficulty of blinding the participants and instructors in exercise trials, only the research staff performing the assessment and statistical analysis were blinded to the exercise group assignment. The allocation concealment method selected was central allocation.

## Statistical methods

Statistical analyses were conducted using SPSS Statistics 23.0 (Armonk, NY, USA). Prior to data analysis, the Kolmogorov–Smirnov test was used to determine the normal distribution of the variables. Levene's test was also performed to determine the homogeneity of variance. Descriptive data are presented as mean ± SD and range. Intention-to-treat analysis using last observation for missing data was conducted. To compare variables before the intervention, analysis of variance for repeated measures was calculated (general linear model). To compare variables after the intervention, ANCOVA analyses with baseline values included as co-variables were used in order to adjust for potential baseline differences in the dependent variables. As additional analyses, Student's $t$-test for dependent samples was used to evaluate variables within groups. The standardized mean differences (Cohen's effect size) between groups (PG, MG and CG) were calculated together with the 95% CIs (*Hopkins et al., 2009*). An effect size (ES) value of 0.20 indicates a small effect, 0.50 indicates a medium effect, and 0.8 indicates a large effect (*Hopkins et al., 2009*). The level of significance was set to $P < 0.05$.

## RESULTS

Figure 1 illustrates the participant flow during the protocol. The period of recruitment was from September to December of 2016. The trial started in January 2016 and ended in May 2016. Table 2 defines the characteristics of the participants at baseline for each group. At the end of the study there were 16 participants in PG, 19 in MG and 14 in CG. The total participation average was of 91.6%.

The main analysis of the present research indicates that there was a significant training × group difference ($P = 0.005$) in the isometric hip extension strength, with PG statistically different ($P = 0.004$) from CG (Table 3). There were no differences between groups regarding isokinetic strength (Table 4) or balance (Table 5).

The additional analysis (intra-group) shows:

a) There was a significant improvement in trunk isometric extension in PG and MG, which was supported by a large effect size (PG: %change = 18.7%, $P = 0.033$, ES = 0.6; MG: %change = 22.2%, $P = 0.019$, ES = 0.82). There was also a significant increase in hip isometric extension in both groups, with a moderate effect size in PG (PG: %change = 35.5%, $P = 0.0003$, ES = 2.06; MG: %change = 21.4%, $P = 0.001$, ES = 0.61) (Table 6).

b) Table 6 shows the isokinetic strength measurements. Hip isokinetic flexion was significantly improved in PG (Hisok_Fl60w: %change = 18.9%, $P = 0.014$, ES = 0.85; Hisok_Fl120w: %change = 18.3%, $P = 0.038$, ES = 0.57) and every hip isokinetic variable was significantly improved in MG (Hisok_Fl60w: %change = 33.1%, $P = 0.000004$, ES = 1.02; Hisok_Fl120w: %change = 33.9%, $P = 0.0001$, ES = 0.95; Hisok_Ex60w: %change = 31.4%, $P = 0.001$, ES = 1.03; Hisok_Ex120w: %change = 26.6%, $P = 0.031$, ES = 0.7).

c) The TUG test results improved significantly in both PG and MG (PG: %change = 4.8%, $P = 0.018$, ES = 0.39; MG: %change = 12.3%, $P = 0.002$, ES = 0.5).

Regarding safety, there were registered adverse events only in CG. The illnesses that caused the four women lost to follow-up in CG were all related to musculoskeletal diseases: two broken wrists after a fall and two sprained ankles. Exercise programs did not produce any adverse event.

## DISCUSSION

The main objective of the present study was to define whether Pilates or traditional resistance training was better at improving trunk strength and balance in older women. After the 18-week intervention, the Pilates group obtained better results than the control group regarding hip isometric extension strength. There were no other statistical differences between groups in the other isometric or isokinetic trunk and hip variables as well as in the static and dynamic balance. As additional results, at the end of the study the Pilates and Muscular groups improved significantly in dynamic balance and trunk and hip isometric extension strength. Moreover, the Pilates group significantly increased the isokinetic hip flexion and the Muscular group significantly increased every isokinetic variable.

The main result of this study is that scores obtained in the Pilates group were statistically greater than the control group regarding hip isometric extension strength, with a difference of 40.82 N/m between groups. In this regard, it should be highlighted that our additional results showed a significant increase in isometric hip extension strength for both the Pilates and Muscular groups but this was not enough to produce significant differences between the Muscular and control group. A possible explanation for this might be that Muscular group showed higher basal values (111.83 ± 47.8 N/m) than the Control Group (106.81 ± 30.3 N/m) or Pilates Group (100.19 ± 19 N/m).

On the other hand, this result could be associated with the training methodology conducted in the Pilates program. Although Pilates and traditional resistance exercise programs contained similar spine, hip and girdle region exercises, stimulating the muscles in a dynamic and static way, in the Pilates exercise program training instructions were always focused on the Pilates principles (*Wells, Kolt & Bialocerkowski, 2012*) and a prone or supine body posture was adopted habitually. The more controlled and accurate movement accomplished in the Pilates group can assist better neural adaptations (i.e., the coordination of muscle recruitment) that could subsequently be transferred to movement control (*Carroll, Riek & Carson, 2001*): following *Carroll, Riek & Carson (2001)*, this

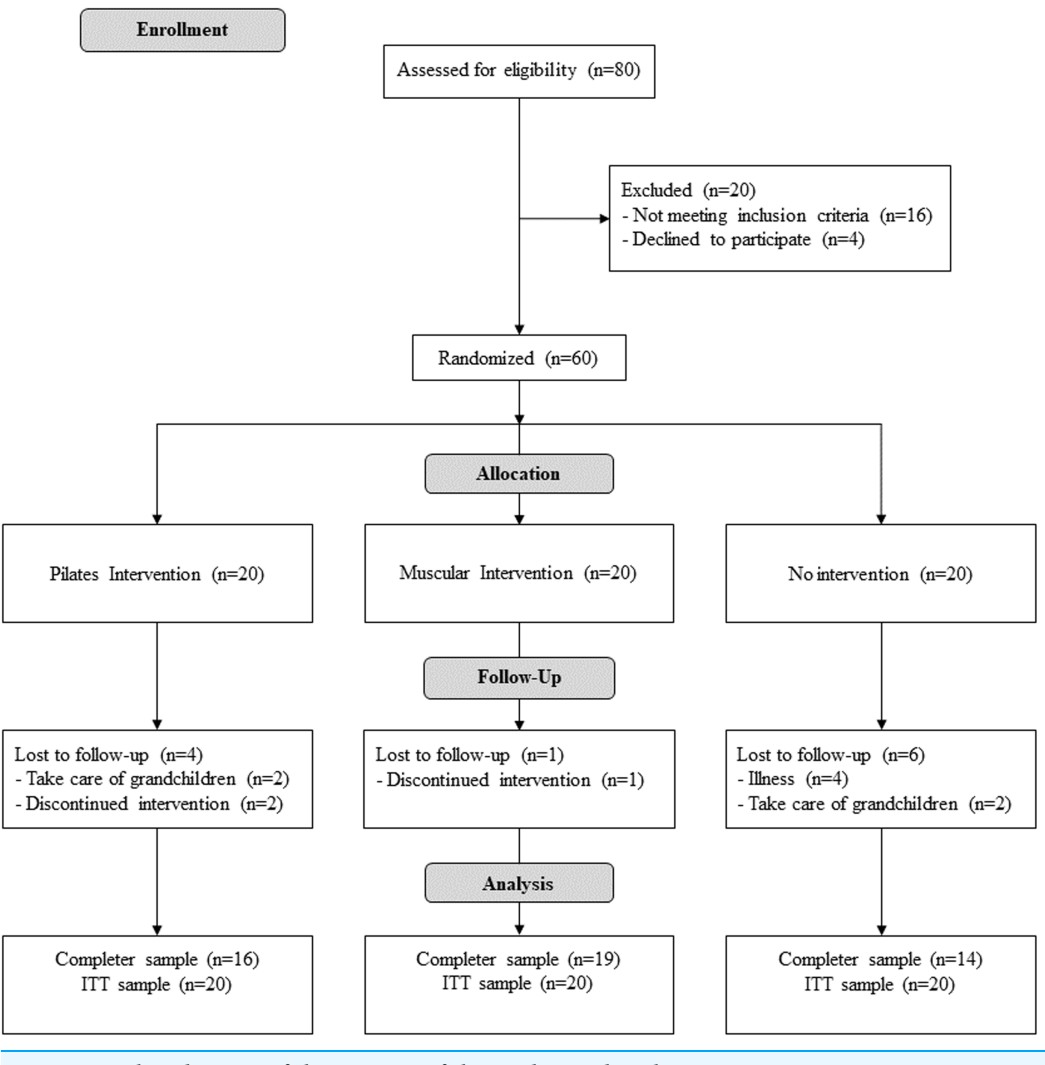

**Figure 1 Flow diagram of the progress of the randomized trial.**

fact and the more frequent body-lying posture could have enhanced the performance in related functional tasks. It can thus be suggested that due to the Pilates specific training methodology, women in the Pilates group showed higher values (larger effect) than women in the Muscular group (moderate effect) regarding isometric hip extension test.

Thus, despite that Pilates exercises entails dynamic exercises, the exercises conducted in the Pilates program entailed greater use of the hip extension muscles in an isometric way, which explains the increased isometric hip extension strength. In the meta-analysis of *De Souza et al. (2017)*, it was pointed out that Pilates is effective for improving strength in older individuals. There were just three studies where core strength was measured (*Katz et al., 1963*; *Meyer et al., 2013*; *Sekendiz et al., 2007*) but hip extension strength was not registered in any case and an isokinetic dynamometer was only used in one of the studies. In the study of *Irez (2014)*, a 14-week exercise program held 3 days per week, 60 min per session, was accomplished in older individuals (aged 65 and over). Two exercise groups

**Table 2 Sample characteristics at baseline (*n* = 60).**

| Variables | *n* | Mean | SD | Min | Max | *P* |
|---|---|---|---|---|---|---|
| Age (years) | | | | | | |
| Pilates | 20 | 67.50 | 3.87 | 62 | 78 | |
| Muscular | 20 | 73.36 | 4.84 | 62 | 80 | 0.000003^" |
| Control | 20 | 65.89 | 4.54 | 60 | 76 | |
| Height (cm) | | | | | | |
| Pilates | 20 | 152.1 | 6.24 | 138.2 | 164.6 | |
| Muscular | 20 | 150.10 | 6.02 | 140 | 164.2 | 0.718 |
| Control | 20 | 154.41 | 6.88 | 140 | 165 | |
| Weight (kg) | | | | | | |
| Pilates | 20 | 74.62 | 11.65 | 56.8 | 94.8 | |
| Muscular | 20 | 71.98 | 11.95 | 53.6 | 101.2 | 0.108 |
| Control | 20 | 72.03 | 11.43 | 51.7 | 99.3 | |
| BMI (kg/m²) | | | | | | |
| Pilates | 20 | 32.32 | 5.24 | 25.38 | 42.42 | |
| Muscular | 20 | 31.95 | 4.84 | 24.86 | 43.88 | 0.576 |
| Control | 20 | 30.54 | 6.36 | 19.46 | 41.12 | |
| SB_time (s) | | | | | | |
| Pilates | 20 | 14.18 | 8.50 | 1 | 30 | |
| Muscular | 20 | 12.96 | 9.84 | 1.38 | 30 | 0.849 |
| Control | 20 | 14.77 | 12.32 | 2.5 | 30 | |
| SB_Vml (mm/s) | | | | | | |
| Pilates | 20 | 3.12 | 2.67 | 0.41 | 9.96 | |
| Muscular | 20 | 2.53 | 2.11 | 0.23 | 7 | 0.585 |
| Control | 20 | 2.34 | 2.56 | 0.18 | 7.7 | |
| SB_Vap (mm/s) | | | | | | |
| Pilates | 20 | 5.11 | 4.67 | 0.79 | 15.64 | |
| Muscular | 20 | 3.83 | 2.61 | 0.25 | 9.24 | 0.485 |
| Control | 20 | 3.86 | 4.04 | 0.2 | 11.85 | |
| SB_Varea (mm/s²) | | | | | | |
| Pilates | 20 | 2.58 | 2.26 | 0.34 | 8.34 | |
| Muscular | 20 | 2.3 | 1.92 | 0.14 | 6.96 | 0.87 |
| Control | 20 | 2.23 | 2.45 | 0.11 | 6.77 | |
| TUG (s) | | | | | | |
| Pilates | 20 | 6.99 | 0.79 | 5.55 | 8.76 | |
| Muscular | 20 | 8.16 | 1.42 | 6.46 | 10.9 | 0.00038*^ |
| Control | 20 | 8.54 | 1.23 | 6.61 | 11.3 | |
| Tisom_Flw (N/m) | | | | | | |
| Pilates | 20 | 198.53 | 78.4 | 51.21 | 365.45 | |
| Muscular | 20 | 234.66 | 67.0 | 125.1 | 368.47 | 0.222 |
| Control | 20 | 231.24 | 70.8 | 95.93 | 415.86 | |

| Variables | n | Mean | SD | Min | Max | P |
|---|---|---|---|---|---|---|
| Tisom_Exw (N/m) | | | | | | |
| Pilates | 20 | 78.96 | 29.0 | 17.84 | 128.79 | |
| Muscular | 20 | 80.1 | 26.7 | 36.97 | 129.41 | 0.723 |
| Control | 20 | 86.64 | 38.6 | 18.57 | 153.38 | |
| Hisom_Flw (N/m) | | | | | | |
| Pilates | 20 | 43.18 | 12.8 | 13.21 | 61.22 | |
| Muscular | 20 | 45.66 | 18.6 | 12.64 | 74.96 | 0.491 |
| Control | 20 | 51.1 | 28.7 | 20.55 | 145.72 | |
| Hisom_Exw N/m | | | | | | |
| Pilates | 20 | 100.19 | 24.6 | 53.28 | 152.75 | |
| Muscular | 20 | 111.83 | 47.8 | 38.18 | 248.06 | 0.586 |
| Control | 20 | 106.81 | 30.3 | 38.37 | 158.67 | |
| Hisok_Fl60w (N/m) | | | | | | |
| Pilates | 20 | 43.94 | 11.3 | 26.3 | 65 | |
| Muscular | 20 | 40.39 | 18.8 | 14.16 | 73.06 | 0.723 |
| Control | 20 | 43.94 | 18.0 | 17.26 | 76.79 | |
| Hisok_Fl120w (N/m) | | | | | | |
| Pilates | 20 | 39.49 | 14.9 | 11.2 | 71.82 | |
| Muscular | 20 | 33.07 | 17.1 | 6.95 | 66.84 | 0.351 |
| Control | 20 | 39.56 | 17.4 | 13.77 | 66.12 | |
| Hisok_Ex60w (N/m) | | | | | | |
| Pilates | 20 | 61.67 | 22.1 | 30.88 | 107.37 | |
| Muscular | 20 | 47.34 | 20.2 | 12.39 | 84.33 | 0.111 |
| Control | 20 | 57.29 | 24.8 | 25.09 | 107.93 | |
| Hisok_Ex120w (N/m) | | | | | | |
| Pilates | 20 | 35 | 18.0 | 10.75 | 72.44 | |
| Muscular | 20 | 35.47 | 17.6 | 8.47 | 79.57 | 0.378 |
| Control | 20 | 43.61 | 27.4 | 10.8 | 127.64 | |

Notes:
SD, Standard Deviation; BMI = kg/m²; SB_time, time maintaining right monopodal static position; SB_Vml, right monopodal displacement velocity in medial-lateral plane; SB_Vap, right monopodal displacement velocity in antero-posterior plane; SB_area, right monopodal velocity moment; TUG, timed up and go; Tisom_Flw, isometric trunk flexion relative to weight; Tisom_Exw, isometric trunk extension relative to weight; Hisom_Flw, isometric hip flexion relative to weight; Hisom_Exw, isometric hip extension relative to weight; Hisok_Fl60w, isokinetic hip flexion at 60°/sg relative to weight; Hisok_Fl120w, isokinetic hip flexion at 120°/sg relative to weight; Hisok_Ex60w, isokinetic hip extension at 60°/sg relative to weight; Hisok_Ex120w, isokinetic hip extension at 120°/sg relative to weight.
[^] $P < 0.05$ differences between Muscular group and Pilates group.
["] $P < 0.05$ differences between Muscular group and Control group.
[*] $P < 0.05$ differences between Control group and Pilates group.

were compared (a Pilates mat group and a walking group) alongside a control group. Isometric hip flexion strength was measured with a manual muscle tester, showing statistical improvement only for the Pilates group. However, differences between groups were not referred to in that study. On the other hand, in the study of *Donath et al. (2016)*, the Pilates group was compared with a multimodal balance training group and a control group. The interventions were conducted over 8 weeks, with two sessions per week, 65 min per session in healthy seniors (75% women; mean age 69.1). In this case, the

**Table 3 Trunk and hip isometric strength parameters.**

| Primary Outcomes | n (ITT) | n (Completer) | Mean of the difference | SD of the difference | ANCOVA interactions (F, P, ES η²) | | | | | | | | |
| | | | | | Training × Group | | | Training × Baseline | | | Training × Age | | |
| | | | | | F | P | ES η² | F | P | ES η² | F | P | ES η² |
| Tisom_Flw N/m | | | | | | | | | | | | | |
| Pilates | 20 | 16 | 24.892 | 89.42 | | | | | | | | | |
| Muscular | 20 | 19 | 9.264 | 55.36 | 0.874 | 0.424 | 0.029 | 3.649 | 0.062 | 0.061 | 1.172 | 0.284 | 0.02 |
| Control | 20 | 14 | 23.797 | 64.65 | | | | | | | | | |
| Tisom_Exw N/m | | | | | | | | | | | | | |
| Pilates | 20 | 16 | 10.227 | 20.02 | | | | | | | | | |
| Muscular | 20 | 19 | 17.094 | 31.85 | 1.24 | 0.297 | 0.041 | 1.358 | 0.249 | 0.023 | 0.901 | 0.247 | 0.015 |
| Control | 20 | 14 | −2.071 | 18.97 | | | | | | | | | |
| Hisom_Flw N/m | | | | | | | | | | | | | |
| Pilates | 20 | 16 | 4.176 | 11.37 | | | | | | | | | |
| Muscular | 20 | 19 | 2.185 | 11.15 | 0.021 | 0.979 | 0.001 | 0.474 | 0.494 | 0.008 | 0.499 | 0.483 | 0.009 |
| Control | 20 | 14 | 2.735 | 6.44 | | | | | | | | | |
| Hisom_Exw N/m | | | | | | | | | | | | | |
| Pilates | 20 | 16 | 41.464 | 44.91 | | | | | | | | | |
| Muscular | 20 | 19 | 19.171 | 24.45 | 5.833 | 0.005 | 0.172 | 0.813 | 0.371 | 0.012 | 0.176 | 0.676 | 0.003 |
| Control | 20 | 14 | 7.815 | 18.36 | | | | | | | | | |

Notes:
Differences between Pilates, Muscular and Control groups.
SD, Standard Deviation; ITT, Intention to treat; Tisom_Flw, isometric trunk flexion relative to weight; Tisom_Exw, isometric trunk extension relative to weight; Hisom_Flw, isometric hip flexion relative to weight; Hisom_Exw, isometric hip extension relative to weight.

balance group was statistically better than the Pilates group regarding isometric trunk extension strength, measured with the modified Sorensen test. However, *Markovic et al. (2015)* did not find any statistical difference in isometric trunk extension strength between a Pilates group, a balance and core resistance training group and a control group after an 8-week program three times per week, 60 min per session in women aged 65–79 years. These results are in accordance with those obtained in the present study regarding trunk strength, but the hip scores are missing again.

It is important to know the prevalence of exercises regarding hip muscle in the Pilates protocols and, to our knowledge, there are no other studies that provide such data. Moreover, from a health-related point of view, hip isometric strength in women declines by an average of 1.31 kg/year between the ages of 70 and 75 years, and 0.39 kg/year thereafter (*Xue et al., 2010*), with faster rates of decline in hip strength predicting mortality (*Xue et al., 2010*). Furthermore, isometric hip strength is associated with the incidence of lower-limb musculoskeletal injuries (*Luedke et al., 2015*), leading to decreased functional status. Isometric hip extension strength is a particular factor that distinguishes fallers from non-fallers (*Gafner et al., 2018*). Consequently, the Pilates exercise program used in the present study could be recommended for promoting daily physical activity development in older women, contributing to diminished risk of falling and a lower risk of dying in older women.

**Table 4 Trunk and hip isokinetic strength parameters.**

| Primary Outcomes | n (ITT) | n (Completer) | Mean of the difference | SD of the difference | ANCOVA interactions (F, P, ES η²) | | | | | | | | |
|---|---|---|---|---|---|---|---|---|---|---|---|---|---|
| | | | | | Training × Group | | | Training × Baseline | | | Training × Age | | |
| | | | | | F | P | ES η² | F | P | ES η² | F | P | ES η² |
| Hisok_Fl60w (N/m) | | | | | | | | | | | | | |
| Pilates | 20 | 16 | 6.705 | 11.23 | | | | | | | | | |
| Muscular | 20 | 19 | 13.786 | 11.5 | 1.015 | 0.369 | 0.035 | 1.149 | 0.288 | 0.02 | 0.301 | 0.585 | 0.005 |
| Control | 20 | 14 | 5.658 | 15.82 | | | | | | | | | |
| Hisok_Fl120w (N/m) | | | | | | | | | | | | | |
| Pilates | 20 | 16 | 5.941 | 11.98 | | | | | | | | | |
| Muscular | 20 | 19 | 12.27 | 12.92 | 17.53 | 0.183 | 0.06 | 0.143 | 0.707 | 0.002 | 0.058 | 0.81 | 0.001 |
| Control | 20 | 14 | 5.444 | 14.04 | | | | | | | | | |
| Hisok_Ex60w (N/m) | | | | | | | | | | | | | |
| Pilates | 20 | 16 | 0.801 | 26.15 | | | | | | | | | |
| Muscular | 20 | 19 | 15.541 | 19.9 | 0.872 | 0.424 | 0.028 | 467.6 | 0.035 | 0.076 | 0.002 | 0.967 | 0 |
| Control | 20 | 14 | 6.965 | 25.95 | | | | | | | | | |
| Hisok_Ex120w (N/m) | | | | | | | | | | | | | |
| Pilates | 20 | 16 | 2.716 | 14.22 | | | | | | | | | |
| Muscular | 20 | 19 | 8.876 | 17.7 | 0.742 | 0.481 | 0.026 | 12.924 | 0.261 | 0.022 | 0.022 | 0.881 | 0 |
| Control | 20 | 14 | 0.336 | 14.88 | | | | | | | | | |

Notes:
Differences between Pilates, Muscular and Control groups.
SD, Standard Deviation; ITT, Intention to treat; Hisok_Fl60w, isokinetic hip flexion at 60°/sg relative to weight; Hisok_Fl120w, isokinetic hip flexion at 120°/sg relative to weight; Hisok_Ex60w, isokinetic hip extension at 60°/sg relative to weight; Hisok_Ex120w, isokinetic hip extension at 120°/sg relative to weight.

Regarding the additional analysis results, there were significant improvements in isometric trunk and hip extension and isokinetic hip flexion strength after the 18-week training period in the Pilates and group. These findings are in accordance with other studies(*Irez et al., 2011*; *Markovic et al., 2015*; *Bertoli et al., 2018*). One unexpected finding was that isokinetic hip extensor strength showed no improvement after the Pilates program. This could indicate that there was a prevalence of exercises based on dynamic hip flexion rather than dynamic hip extension in the Pilates program. Dynamic hip extensions can only be conducted in prone or four-footed positions, which are more complex for older women to adopt. This may lead to a lack of prone or four-footed position exercises in the Pilates sessions, which should be addressed in Pilates protocols in order to avoid muscular imbalance.

For its part, Muscular program participants significantly increased either their trunk and hip isometric extension or the trunk and hip isokinetic strength at 60°/s and 120°/s, which was accompanied by a moderate to high effect sizes. The large increase in the Muscular group could be attributed to greater neural mechanisms, as the exercises more frequently involved other parts of the body (i.e., upper or lower extremities). It is well known that strength training can assist neural adaptations (i.e., the coordination of muscle recruitment), which could subsequently be transferred to movement control (*Carroll, Riek & Carson, 2001*). Traditionally, mobility, balance and functionality impairments in old people has been associated to aged-related lower extremities changes (*Fukagawa et al.,*

**Table 5 Static and dynamic balance parameters.**

| Secondary Outcomes | n (ITT) | n (Completer) | Mean of the difference | SD of the difference | ANCOVA interactions (*F*, *p*, ES η²) | | | | | | | | |
| --- | --- | --- | --- | --- | --- | --- | --- | --- | --- | --- | --- | --- | --- |
| | | | | | Training × Group | | | Training × Baseline | | | Training × Age | | |
| | | | | | *F* | *P* | ES η² | *F* | *P* | ES η² | *F* | *P* | ES η² |
| SB_time (s) | | | | | | | | | | | | | |
| Pilates | 20 | 16 | 0.501 | 10.87 | | | | | | | | | |
| Muscular | 20 | 19 | 1.824 | 8.06 | 1.73 | 0.187 | 0.041 | 18.33 | 0.001 | 0.217 | 7.7 | 0.008 | 0.091 |
| Control | 20 | 14 | 1.121 | 4.08 | | | | | | | | | |
| SB_Vml (mm/s) | | | | | | | | | | | | | |
| Pilates | 20 | 16 | −0.496 | 3.40 | | | | | | | | | |
| Muscular | 20 | 19 | 0.102 | 1.44 | 0.546 | 0.582 | 0.012 | 27.356 | <0.001 | 0.306 | 5.992 | 0.018 | 0.067 |
| Control | 20 | 14 | −0.104 | 1.5 | | | | | | | | | |
| SB_Vap (mm/s) | | | | | | | | | | | | | |
| Pilates | 20 | 16 | −0.541 | 5.71 | | | | | | | | | |
| Muscular | 20 | 19 | 0.91 | 3.11 | 0.38 | 0.686 | 0.009 | 27.466 | <0.001 | 0.311 | 5.171 | 0.027 | 0.059 |
| Control | 20 | 14 | 0.302 | 2.59 | | | | | | | | | |
| SB_Varea (mm/s²) | | | | | | | | | | | | | |
| Pilates | 20 | 16 | −0.215 | 2.7 | | | | | | | | | |
| Muscular | 20 | 19 | 0.433 | 1.86 | 0.086 | 0.917 | 0.002 | 237.979 | <0.001 | 0.282 | 55.158 | 0.022 | 0.065 |
| Control | 20 | 14 | −0.19 | 1.22 | | | | | | | | | |
| TUG (s) | | | | | | | | | | | | | |
| Pilates | 20 | 16 | −0.261 | 0.46 | | | | | | | | | |
| Muscular | 20 | 19 | −0.677 | 0.87 | 2.359 | 0.104 | 0.067 | 9.798 | 0.003 | 0.140 | 0.5 | 0.482 | 0.007 |
| Control | 20 | 14 | −0.301 | 0.68 | | | | | | | | | |

**Notes:**
Differences between Pilates, Muscular and Control groups.
SD, Standard Deviation; ITT, Intention to treat; SB_time, time maintaining right monopodal static position; SB_Vml, right monopodal displacement velocity in medial-lateral plane; SB_Vap, right monopodal displacement velocity in antero-posterior plane; SB_area, right monopodal velocity moment; TUG, timed up and go.

*1995*). Nevertheless, trunk stability and strength could enhance old people mobility and functionality, favoring the development of daily physical activities and reducing the risk of falling (*Granacher et al., 2013*). In this regard, *Irez et al. (2011)* showed significant changes in dynamic balance, the sit and reach test, muscle strength and a decreased risk of falling when integrating Pilates into an exercise program using elastic resistance bands in older women. Hence combining the Muscular and the Pilates programs could increase the functional performance and quality of life in older women.

Regarding SB, and against our hypothesis, no changes were found in any of the experimental groups after training and no differences were found between groups. In contrast, *Bird, Hill & Fell (2012)* showed changes in static and dynamic balance following 5 weeks of Pilates training. *Kibar et al. (2016)*, observed that an 8-week Pilates training program could improve SB, flexibility, abdominal muscle endurance, and abdominal and lumbar muscle activity. In addition, strength training may increase balance in older people (*Lee & Park, 2013*). In this way, a previous systematic review (*Orr, Raymond & Fiatarone Singh, 2008*) concluded that the inconsistent effect of the resistance training programs on balance may be explained by several factors: the

**Table 6 Trunk and hip isometric and isokinetic strength parameters pre- and post- intervention in Pilates, Muscular and Control groups.**

| Variables | Pre-training | | | Post-training | | | P | 95% CI for mean difference | | Cohen's d |
|---|---|---|---|---|---|---|---|---|---|---|
| | n | Mean | SD | n | Mean | SD | | Lower | Upper | |
| Tisom_Flw N/m | | | | | | | | | | |
| Pilates | 20 | 198.53 | 78.4 | 16 | 251 | 91.3 | 0.231 | −84.19 | 21.96 | 0.64 |
| Muscular | 20 | 234.66 | 67.0 | 19 | 245.97 | 72.0 | 0.441 | −39.48 | 18.03 | 0.19 |
| Control | 20 | 231.24 | 70.8 | 14 | 272.65 | 109.3 | 0.14 | −72.39 | 11.2 | 0.91 |
| Tisom_Exw N/m | | | | | | | | | | |
| Pilates | 20 | 78.96 | 29.0 | 16 | 97.14 | 31.3 | 0.033 | −24.37 | −1.19 | 0.60 |
| Muscular | 20 | 80.1 | 26.7 | 19 | 100.77 | 43.0 | 0.019 | −35.98 | −3.61 | 0.82 |
| Control | 20 | 86.64 | 38.6 | 14 | 84.97 | 34.2 | 0.653 | −9.84 | 15.17 | 0.23 |
| Hisom_Flw N/m | | | | | | | | | | |
| Pilates | 20 | 43.18 | 12.8 | 16 | 50.4 | 15.3 | 0.117 | −11.92 | 1.48 | 0.51 |
| Muscular | 20 | 45.66 | 18.6 | 19 | 48.48 | 20.5 | 0.37 | −8.31 | 3.25 | 0.25 |
| Control | 20 | 51.1 | 28.7 | 14 | 55.76 | 32.0 | 0.089 | −7.65 | 0.62 | 0.01 |
| Hisom_Exw N/m | | | | | | | | | | |
| Pilates | 20 | 100.19 | 24.6 | 16 | 153.54 | 50.4 | 0.0003 | −75.55 | −28.11 | 2.06 |
| Muscular | 20 | 111.83 | 47.8 | 19 | 136.79 | 57.4 | 0.001 | −34.26 | −10.13 | 0.61 |
| Control | 20 | 106.81 | 30.3 | 14 | 112.72 | 44.2 | 0.088 | −21.83 | 1.74 | 0.50 |
| Hisok_Fl60w N/m | | | | | | | | | | |
| Pilates | 20 | 43.94 | 11.3 | 16 | 54.17 | 17,0 | 0.014 | −14.79 | −1.97 | 0.85 |
| Muscular | 20 | 40.39 | 18.8 | 19 | 58.09 | 21,1 | 0.000004 | −21.19 | −10.73 | 1.02 |
| Control | 20 | 43.94 | 18.0 | 14 | 50.63 | 18,4 | 0.149 | −17.51 | 2.96 | 0.58 |
| Hisok_Fl120w N/m | | | | | | | | | | |
| Pilates | 20 | 39.49 | 14.9 | 16 | 48.35 | 18.5 | 0.038 | −14.37 | −0.48 | 0.57 |
| Muscular | 20 | 33.07 | 17.1 | 19 | 47.88 | 16.7 | 0.0001 | −20.41 | −8 | 0.95 |
| Control | 20 | 39.56 | 17.4 | 14 | 45.15 | 19.9 | 0.119 | −10.26 | 9.39 | 0.54 |
| Hisok_Ex60w N/m | | | | | | | | | | |
| Pilates | 20 | 61.67 | 22.1 | 16 | 63.67 | 38.1 | 0.893 | −16.68 | 14.68 | 0.09 |
| Muscular | 20 | 47.34 | 20.2 | 19 | 46.45 | 23.6 | 0.001 | −27.82 | −8.17 | 1.03 |
| Control | 20 | 57.29 | 24.8 | 14 | 62.99 | 37.9 | 0.274 | −25.9 | 7.99 | 0.39 |
| Hisok_Ex120w N/m | | | | | | | | | | |
| Pilates | 20 | 35 | 18.0 | 16 | 39.71 | 21.7 | 0.407 | −11.88 | 5.09 | 0.25 |
| Muscular | 20 | 35.47 | 17.6 | 19 | 48.34 | 22.8 | 0.031 | −19.3 | −1.26 | 0.70 |
| Control | 20 | 43.61 | 27.4 | 14 | 37.75 | 21.0 | 0.926 | −10.26 | 9.4 | 0.19 |

**Note:**

SD, Standard Deviation; PG, Pilates Group; MG, Muscular Group; CG, Control Group; Hisok_Fl60w, isokinetic hip flexion at 60°/sg relative to weight; Hisok_Fl120w, isokinetic hip flexion at 120°/sg relative to weight; Hisok_Ex60w, isokinetic hip extension at 60°/sg relative to weight; Hisok_Ex120w, isokinetic hip extension at 120°/sg relative to weight.

heterogeneity of cohort and balance tests, the variability in methodology of the balance test and the sample size, the inadequate dose of resistance training and/or compliance to training, the lack of statistical power, and that strength training alone is not robust enough to improve balance.

However, our results showed a significant improvement in the TUG test in both the Pilates and the Muscular groups. These results are in line with previous Pilates

(*Bergamin et al., 2015*; *Bird et al., 2009*; *Kaesler et al., 2007*; *Mokhtari, Nezakatalhossaini & Esfarjani, 2013*) and traditional resistance training programs (*Marques et al., 2011*; *Kang et al., 2015*). One possible explanation for these dynamic balance improvements may be the increase in lower limb and abdominal strength and the improved postural control (*Bergamin et al., 2015*). Pilates exercises are based on movement control, which can lead to changes in the nervous system through alterations of synaptic connections and cortical remapping (*Bolognini, Pascual-Leone & Fregni, 2009*). Pilates can also improve core stability and make an individual more kinesthetically aware of how to reduce faulty movement patterns (*Johnson et al., 2007*), thus resulting in improved motor control. In addition, a previous systematic review (*Cadore et al., 2013*) regarding different exercise intervention showed that the TUG improved after the strength training period with an increment of 7.2–40%. It was associated with increased strength in the lower limbs and abdominal muscles and optimized postural control (*Bolognini, Pascual-Leone & Fregni, 2009*). Ours results suggest that Pilates training and resistance training were effective to increase the mobility in older women and may contribute to diminished fall rates.

The clinical implications of the present study are related to the hip muscle enhancement that comes with Pilates training. Practicing Pilates twice a week (1 h per session) for 18 weeks in a moderate-to-vigorous intensity and increasing resistance with elastic bands controls age-related muscular decline and the associated lower-limb musculoskeletal injuries contributing to the risk of falling. This will also contribute to reduce the health care system spending. In this way, the Pilates program could be recommended by the sanitary, physiotherapist and sports personnel for improving hip strength and for diminishing the risk of falling in aged women. In addition, both training programs showed a trend forward to improve functional and strength variables when compared to the control group. On the other hand, these results should be considered with several limitations. The non-blinding of participants and instructors affects the internal validity. The external validity of the results could not be generalized because of the small sample size at the end of the study. Controlling cognitive function or opening the age range could determine any interaction regarding the results. Moreover, to follow more closely the exercises execution in order to improve the quality performance and to check more frequently the working load adaptation of every participant should be taken into account in order to increase the exercise programs strength and balance effects. Additionally, the number of flexion and extension-based exercises should be controlled in order to avoid muscle imbalance.

## CONCLUSIONS

According to the results obtained in the present study, the Pilates training program seems to be more effective for improving isometric hip and trunk extension strength, and the Muscular training program appear to have greater effects on trunk and hip isokinetic strength, with no significant effects between groups. Additionally, both training programs showed moderate effects for the TUG. Nonetheless, studies with larger sample sizes and longer duration are necessary to clarify the effects of each of the trainings programs used.

## ACKNOWLEDGEMENTS

This research was edited and proofreaded by Proof-Reading-Service.com (United Kingdom).

### Funding

This work was supported by the San Antonio Catholic University (PMAFI/24/14). The funders had no role in study design, data collection and analysis, decision to publish, or preparation of the manuscript.

### Grant Disclosures

The following grant information was disclosed by the authors:
San Antonio Catholic University: PMAFI/24/14.

### Competing Interests

The authors declare that they have no competing interests.

### Author Contributions

- María Carrasco-Poyatos conceived and designed the experiments, performed the experiments, analyzed the data, contributed reagents/materials/analysis tools, prepared figures and/or tables, authored or reviewed drafts of the paper, approved the final draft.
- Domingo J. Ramos-Campo analyzed the data, prepared figures and/or tables, authored or reviewed drafts of the paper, approved the final draft, search journals for publication.
- Jacobo A. Rubio-Arias analyzed the data, contributed reagents/materials/analysis tools, prepared figures and/or tables, approved the final draft.

### Human Ethics

The following information was supplied relating to ethical approvals (i.e., approving body and any reference numbers):

The trial was approved by the San Antonio Catholic University (UCAM) ethics committee.

### Clinical Trial Ethics

The following information was supplied relating to ethical approvals (i.e., approving body and any reference numbers):

The San Antonio Catholic University (UCAM) ethics committee.

### Clinical Trial Registration

The following information was supplied regarding Clinical Trial registration:
NCT02506491

Trial registration: ClinicalTrials.gov (identifier: NCT02506491; available from https://clinicaltrials.gov/show/NCT02506491).

## Data Availability

Raw data is available at Internet Archive (https://archive.org/details/datasetcarrascopoyatos) and as a Supplemental File.

## Supplemental Information

Supplemental information for this article can be found online at http://dx.doi.org/10.7717/peerj.7948#supplemental-information.

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
