# Peer review of "Pilates versus resistance training on trunk strength and balance adaptations in older women: a randomized controlled trial"

_PeerJ, doi:10.7717/peerj.7948_

## Round 0.1 · original submission · Major Revisions

The three reviewers and I see some potential in this study in the manuscript, but have highlighted a number of areas that you need to consider if you wish the manuscript to be accepted for publication.

Reviewer 1 ·

Basic reporting

Spelling and grammatical errors throughout manuscript- suggest re-read of manuscript from someone external to the research team. For example in the abstract, starting a sentence with PG is not recommended.
Inconsistency of word "program" and "programmes" and "elderly" and "older" women

Line 46, 98, 100,: reword sentence
Line 51: New sentence "Thus..."
Line 53; unsure what the .. are for?
Line 54, 259, 312, 326: insert reference
Line 60; provide more than the one reference you have as you included "studies"

Line 89: provide ethics number
Line 107: include the number of sessions with %

Throughout the manuscript, you need to be consistent on what one of the groups is, muscular group or traditional resistance training? very confusing?

Line 234: included results to strengthen your statement in the discussion

Line 257; two or three? you include three studies after this sentence, recheck? You should also include more detail about these studies age bracket and whether this related to females or both genders included int he results for the comparison of the results you have for this manuscript.

Inconsistency of referencing formatting

Figure 1: consistency of either 'intervention" or "group" for the manuscript.
Include what ITT stands for
Curious as to why participants discontinued intervention for Pilates and muscular intervention
All tables: rephrase titles as they should stand without the manuscript.
Table 1: include weeks 1-2 in table
Table 2-7: include key legend of words missing eg. SD missing what * for p-values represent
Table 4 -5: what does * mean?

Experimental design

I think you need to provide more detail and make the link very clear with your research topic as to why isometric and isokinetic strength is important to measure (in particular in the introduction section).You also need to include something about TUG in the introduction as you don't mention this in the introduction but talk about it in the results and discussion section. Suggest changing the title as well as it doesn't link to the manuscript.

Lines 91-92: include ethics number

Line 100: what did the medical evaluation entail to rule out physical and mental restrictions that were included in your exclusion criteria

In regards to inclusion criteria, why do you restrict women above 81 years of age? and why do you restrict women who had cognitive impairment?

Outcomes section needs to be separated as you also include protocol information in here and this doesn't really fit under the heading. You will also need to include references for your measurement protocols to justify why you selected those measurements and validation for the population you have researched. You will also need to include more detail on the acronyms provided in this section eg.line 155 /sg???

May you please provide clarification on lines 203-204 as you had the recruitment until December but previously stated that your study was restructured, just a bit confusing to read.
Line 206: was this percentage weekly? total? provide detail.
Line 210: include P-values

Include more detail on the study limitations.

I believe there are too many tables for this manuscript - 7 tables are just too much.

Validity of the findings

Line 206: why were registered adverse events only in CG and not in the two groups that were of particular focus for the research topic?
Line 229: what was defined as adverse events?

Additional comments

I suggest making the changes recommended to strengthen the manuscript.

Reviewer 2 ·

Basic reporting

The manuscript was designed to determine whether Pilates or resistance training is better at improving muscular strength (trunk and lower limb) and balance (static and dynamic). Although being used since the beginning of the last century, the benefits of Pilate’s method are still under investigation. So, RCTs are very welcome and necessary to establish its real benefits, mainly for older adults. As I'm not an English speaker, I'm not considered myself able to raise criticisms about the English style. Besides the relevance of the study, there are some points that must be reviewed by the authors.

Introduction
1) There are some misconceptions that need to be rewritten. First of all, I challenge authors to sustain the definition of health (absence of disease and a good physical condition) stated in the manuscript, mainly for older adults. Recently, WHO published a report in that the definition of health aging goes in a different way.

2) The authors must be aware of generalization. In the gerontology field, the optimal achievement of activities of daily living is dependent on physical, mental and environmental factors. So, I suggest that the author rephrase the relationship among core/trunk muscles, balance and activities of daily living (please, exemplified using relevant data/paper published).

3) As the own name says, multicomponent training or multimodal training is considered when two or more of the following components are involved in the physical exercise program: muscular resistance, endurance/walking, balance or flexibility. A training program that only involves muscular resistance/strength exercise, either in machines or elastic bands, cannot be called as multicomponent training.

4) The rationale for support the hypothesis must be improved. It is not clear the differences between the proposed exercises in improving muscular strength and balance. In addition, there is evidence that Pilates improves dynamic balance, but not the static balance. Please, give a scientific basement to support the study’s hypothesis.

Experimental design

The aims and scope are original and relevant. However, the research question must be well defined, as mentioned above. My comments about the material, methods, and experimental design are pointed below.

1) The authors assume that participants are physically and mentally able to participate in the exercise programs, which are attested by a GP. Are the volunteers submitted to tests to confirm these?

2) In addition, there are some inclusion criteria that need more details. Which test and/or scales were used to evaluate activities of daily living (KATZ? LAWTON?), cognitive function (MMSE?) and chronic diseases (self-report?).

3) Please, provide the number of Ethics Board committee approval.

4) Please, make clear in the text if the Pilates exercises were performed on the mat and/or on the apparatus. In addition, provide an illustrated (photo) of each Pilates and resistance training exercise presented in Table 1 (can be included as supplementary data, as well). The breathing pattern described in Table 1 is not clear. There are many Pilates schools in the world, justifying the importance of make the protocol as clear as possible to be replicated.

5) Please, clarify if you are measuring isometric and isokinetic flexion-extension muscle strength of the trunk or of the core. Which muscles are considered being part of the core, by the way?

6) Please, make clear in the text that you are measuring the peak torque (N/m).

7) For both trunk and lower limb, the isokinetic peak torque was evaluated at 60o/s and 120o/s. Why these velocities were chosen? Are they safe for older adults? Are the peak torques presented in Table 2 the average or the greatest of 5 contractions? Is the coefficient of variance of each trial considered for the peak torque calculation (either average or maximal peak torque)?

8) In the postural stability measurement, give more information about the force platform used (example, sampling rate) and how variables were calculated (Matlab code? Specific force platform report? Other software?).

9) If I’m not wrong, the force platform evaluates the center of pressure (COP) displacement, right? If yes, make it clear in the text.

10) Please, explain why only velocity related measures were presented for single leg support. Why COP displacement (RMS) and Area (mm2) were not presented, as they are common measurement calculates from force platforms?

11) Which covariables were considered in ANCOVA analysis?

12) Please, provide more clear information about Cohen’s effect size values (i.e, value ranges for small, medium and large effects).

Validity of the findings

The main finding of the study was the interaction effect (group vs. time) for isometric hip extension (Pilates group showed greater peak torque than the control group). No additional effects were observed. In an intragroup analysis, training protocols showed moderate/large effects in isometric and isokinetic peak torque. In order to improve the understanding, I suggest that the authors present results of the trunk and hip isometric peak torques separately (lines 212 to 216). Additionally, the content of tables 6 and 7 can be presented together (just one table).

In a general analysis, the effects of Pilates training was great for isometric peak torque (both trunk and hip, independently of the movement direction), while the effects of muscular resistance training were great for isokinetic peak torque (both hip flexion and extension, independently of the angular velocity). Pilates training presented insignificant/small effects in isokinetic hip extension and flexion peak torque (independently of angular velocity). As groups were different at baseline for age and TUG performance, authors must make themselves clear about the adjustment of these differences in the ANCOVA (Did age is set as covariable for all analysis? Did TUG was set as covariable, at least for the mobility analysis?)

In relation to secondary outcomes, only mobility improved after exercise training (small effects for Pilates and medium effects for muscular resistance). No changes were observed for static balance.

The discussion section must be improved. Besides some misconceptions (example, in line 313 about the muscular resistance training used being considered as a multicomponent training by the authors), evidence of the importance of trunk and hip strength for balance and functional capacity need to be explored. Also, isometric contraction is very common in Pilates exercises and this must be pointed in the discussion. Even being a secondary outcome, is important to explain the absence of results in the static balance after the exercise training protocols studied.

I recommend that the author rewrite the conclusion. I disagree with the statement that Pilates and muscular resistance exercise training brought similar results to strength variables analyzed. As described above, Pilates presented insignificant/small effects for isokinetic hip extension peak torque, while the effect of muscular resistance on isokinetic hip peak torques was medium/large. In addition, there are no data in the manuscript that support Pilates contribution to a reduced risk of falling and dying in older women.

Additional comments

I have nothing more to add.

·

Basic reporting

The primary and the secondary outcomes are well defined in the text and they match with the ones described in the clinicaltrials.gov protocol. But Cochrane suggest that primary outcomes should be not much. I encourage the authors to try to diminish the number of primary outcomes.

Experimental design

Randomized and controlled clinical trials should present possible adverse events related to the interventions.

Validity of the findings

The clinical implications should be extended. It is necessary to give more evidence on how the results impact on professional practice.

Additional comments

The quality of the RCT is high enough as it follows closely the CONSORT statement. Also, the authors added the CONSORT checklist as a supplement file. In relation to this, I would like the authors to explain why they have conducted an intention-to-treat (ITT) design.

---

## Round 0.2 · Minor Revisions

The authors have addressed many of the comments raised by the reviewers with the submission. Please look to take on board these final comments so that it can be accepted for publication.

Reviewer 2 ·

Basic reporting

The authors did a great job in the manuscript and most of the suggestions were incorporated at the new version of it. Some minor comments are pointed below:

1) lines 39 - 41: Check the abstract conclusion (seems the same as the first version. Please, adjust for the current manuscript conclusion);

2) line 61: Again, be careful with generalizations!!! Reference 11 included postmenopausal older women with osteoporosis (make this clear in the introduction);

3) lines 61-63: the phrase needs to be improved. Suggestion: "Thus, the measurement of isokinetic and isometric hip and trunk strength can offer important information about physical factors related to healthy aging".”

5) lines 66: I still think that the multicomponent training assumption is not clear for the reader. As is written, it seems more machine resistance training than multicomponent training itself. As commented before, multicomponent training includes two or more of the following exercises: muscle resistance/strength, walking/endurance, balance and/or flexibility. I recommend the inclusion of this information that can help to support the idea that Pilates is a multicomponent training regime (as the authors pointed in the answer file for the reviewers). In addition, I think that it is not clear that balance training was included in both protocols (mainly in the muscular group). If I’m not wrong, a program is considered multicomponent if all components (in the present case, muscle resistance and balance) have the same weight in the training (duration and/or volume).

6) lines 66-69: The cited reference (number 12) is about Pilates training and not about muscle resistance training. The phrase seems to be a little bit out of the context which is justified due to the confusion with the definition of multicomponent training. The authors must make themselves more clear and avoid generalizations. As this phrase is related to Pilates, I believe that it is more adequate to change its position (after line 79) or suppress it.

7) line 73: What do the authors mean about dual-task changes? I checked reference number 15 and nothing is mentioned about the benefits of Pilates training in dual-task performance. Please, review this phrase.

8) line 177: displacement velocity of what? Center of pressure (COP), I guess. The authors affirm that this information was included in the new version of the manuscript (in the answer file for the reviewers), but I couldn't find it.

9) line 179: What is SB (single leg balance? static balance?)? Please, make all abbreviations clear.

10) line 182: Please, specify which TUG velocity (usual or fast) was performed.

11) line 183: Please, modify the phrase “The best time was recorded” to “The best time was used in the analyzes”.

12) line 282: I think that the word “manual” is missing (ie., manual muscle tester).

13) lines 293-294: the phrase is a little bit confusing. I didn't understand what the authors mean with “the incidence of exercise regarding hip muscles” (would it be: the prevalence of exercises regarding hip muscle in the Pilates protocols?).

14) line 319: The number of the cited reference is missing.

15) line 348: Did the authors exclude reference number 54? If yes, it must be suppressed from the final reference list.

Experimental design

Nothing to add

Validity of the findings

Nothing to add

Additional comments

Nothing to add

·

Basic reporting

I agree with the text.

Experimental design

I agree with the text.

Validity of the findings

I agree with the text.

---

## Round 0.3 · Minor Revisions

The paper is acceptable for publication pending the last comment regarding dual tasking by the reviewer.

Reviewer 2 ·

Basic reporting

not to add

Experimental design

not to add

Validity of the findings

not to add

Additional comments

The authors responded to all the questions made. I just have one last commentary. Dual-task performance refers to the ability to conduct two tasks simultaneously, such as walking while talking on the phone or while holding a tray. The transference of the benefits of exercise training to the dual-task performance itself is very difficult. As far as I know, we don't have papers that support the benefit of Pilates in dual-task performance. I let the authors decide if they want to keep the following phrase: "Furthermore, during recent years a new type of training program called Pilates has been included as an effective method for enhancing dual-task changes". If yes, they need to change the reference (Wells et al., 2012), once this review paper did not support the effectiveness of Pilates on dual-task performance.

---

## Round 0.4 · accepted · Accept

I thank the authors for taking on board all the comments from the reviewers and I. I am therefore happy to recommend this manuscript except for publication in PeerJ.